

# Seasonal characteristics of organic aerosol chemical composition and volatility in Stuttgart, Germany

Wei Huang[1,2], Harald Saathoff[1], Xiaoli Shen[1,2], Ramakrishna Ramisetty[1,3], Thomas Leisner[1,4], Claudia Mohr[5,*]

[1]Institute of Meteorology and Climate Research, Karlsruhe Institute of Technology, Eggenstein-Leopoldshafen, 76344, Germany

[2]Institute of Geography and Geoecology, Working Group for Environmental Mineralogy and Environmental System Analysis, Karlsruhe Institute of Technology, Karlsruhe, 76131, Germany

[3]Now at: TSI Instruments India Private Limited, Bangalore, 560102, India

[4]Institute of Environmental Physics, Heidelberg University, Heidelberg, 69120, Germany

[5]Department of Environmental Science and Analytical Chemistry, Stockholm University, Stockholm, 11418, Sweden

*Correspondence to: claudia.mohr@aces.su.se

**Abstract.** Chemical composition and volatility of organic aerosol (OA) particles were investigated during July–August 2017 and February–March 2018 in the city of Stuttgart, one of the most polluted cities in Germany. Total non-refractory particle mass was measured with a high-resolution time-of-flight aerosol mass spectrometer (HR-ToF-AMS; hereafter AMS). Aerosol particles were collected on filters and analyzed in the laboratory with a filter inlet for gases and aerosols coupled to a high-resolution time-of-flight chemical ionization mass spectrometer (FIGAERO-HR-ToF-CIMS; hereafter CIMS), yielding the molecular composition of oxygenated OA (OOA) compounds. While the average organic mass loadings are lower in the summer period ($5.1 \pm 3.2$ µg m$^{-3}$) than in the winter period ($8.4 \pm 5.6$ µg m$^{-3}$), we find relatively larger mass contributions of organics measured by AMS in summer ($68.8 \pm 13.4$ %) compared to winter ($34.8 \pm 9.5$ %). CIMS mass spectra show OOA compounds in summer have O:C ratios of $0.82 \pm 0.02$ and are more influenced by biogenic emissions, while OOA compounds in winter have O:C ratios of $0.89 \pm 0.06$ and are more influenced by biomass burning emissions. Volatility parametrization analysis shows that OOA in winter is less volatile with higher contributions of low volatile organic compounds (LVOC) and extremely low volatile organic compounds (ELVOC). We partially explain this by the higher contributions of compounds with shorter carbon chain lengths and higher number of oxygen atoms, i.e. higher O:C ratios in winter. Organic compounds desorbing from the particles deposited on the filter samples also exhibit a shift of signal to higher desorption temperatures (i.e. lower apparent volatility) in winter. This is consistent with the relatively higher O:C ratios in winter, but may also be related to higher particle viscosity due to the higher contributions of larger molecular-weight LVOC and ELVOC, interactions between different species and/or particles (particle matrix), and/or thermal decomposition of larger molecules. The results suggest that whereas lower temperature in winter may lead to increased partitioning of semi-volatile organic compounds (SVOC) into the particle phase, this does not result in a higher overall volatility of OOA in winter, and that the difference in



sources and/or chemistry between the seasons plays a more important role. Our study provides insights into the

seasonal variation of molecular composition and volatility of ambient OA particles, and into their potential sources.

## 1 Introduction

Air pollution has significant impacts on human health (D'Amato et al., 2014), visibility (Majewski et al., 2014),

and also interacts with climate change (Seinfeld and Pandis, 2016). Due to rapid urbanization, industrialization,

and growing human population, air quality in urban environments has become a severe issue in more and more

cities all over the world, particularly in densely populated megacities (Guttikunda et al., 2014; Chan and Yao,

2008; Mayer, 1999; Marlier et al., 2016). Air quality in urban environments is influenced by emissions, e.g. from

sources such as industrial processes, automobile traffic, and domestic heating, and also by meteorological

conditions (e.g. solar radiation, wind, temperature, precipitation), atmospheric dispersion, chemical

transformation, location, and topography (D'Amato et al., 2014; Baumbach and Vogt, 2003; Kinney, 2018).

Moreover, air pollution is not limited within the boundaries of urban areas, but can be transported over long

distances and contribute to background pollution on the regional to global scale (Baklanov et al., 2016).

The most abundant air pollutants are nitrogen dioxide ($NO_2$), ozone ($O_3$), sulfur dioxide ($SO_2$), and particulate

matter (PM; D'Amato et al., 2014). Despite its abundance and important impacts on climate and health, PM

sources, physicochemical transformation, and fate in the atmosphere still remain to be fully understood in urban

areas. This is especially true for the organic fraction (Hallquist et al., 2009; Fuzzi et al., 2015). Organic aerosol

(OA) often makes up a significant fraction of submicron particulate mass in urban areas (Hallquist et al., 2009;

Jimenez et al., 2009). OA can be directly emitted into the atmosphere from sources such as fossil fuel combustion

and biomass burning (primary organic aerosol, POA), or be formed in the atmosphere from the oxidation of gas-

phase precursors (secondary organic aerosol, SOA). POA is dominated by vehicular emissions in urban

environments (Bhattu, 2018). SOA, which dominates the global budget of OA (Shrivastava et al., 2015), can be of

biogenic and/or anthropogenic origin with biogenic sources dominating on a global scale (Heald et al., 2008). SOA

also generally makes up the biggest mass fraction of OA in urban areas, as observed e.g. in Mexico City (Volkamer

et al., 2006; Kleinman et al., 2008), some heavily urbanized areas in the U.S. (de Gouw et al., 2005; Zhang et al.,

2005), and during the severe haze pollution events in the big cities in China (Huang et al., 2014). In European

cities, most of the OA mass observed consists of oxygenated compounds (oxygenated organic aerosol, OOA), and

most OOA is of secondary origin and thus SOA (Lanz et al., 2007; Jimenez et al., 2009; Zhang et al., 2007; El

Haddad et al., 2013). Robinson et al. (2007) suggested that semi-volatile and intermediate VOC may play a

dominant role in SOA formation in urban locations. In order to design effective mitigation strategies for urban air

pollution, it is therefore of great importance to identify the sources of OA, and especially SOA, in urban areas.

Source apportionment of OA has been advanced by the application of positive matrix factorization (PMF) to

aerosol mass spectrometer (AMS) or aerosol chemical speciation monitor (ACSM) data (Canonaco et al., 2015;

Crippa et al., 2014; Ulbrich et al., 2009). However, the distinction of OOA sources (biogenic or anthropogenic) by

AMS- or ACSM-PMF remains difficult due to excessive fragmentation and thus loss of molecular information in

the AMS or ACSM. The recent advent of new methods provides more insights into the molecular composition of

OA, such as linear trap quadrupole (LTQ) Orbitrap mass spectrometry (Daellenbach et al., 2019), filter inlet for

gases and aerosols coupled to a high-resolution time-of-flight chemical ionization mass spectrometer (FIGAERO-

HR-ToF-CIMS; Huang et al., 2019), and the newly developed extractive electrospray ionization time-of-flight



mass spectrometer (EESI-ToF; Qi et al., 2019). Whereas AMS-/ACSM-PMF is not directly able to reveal OOA

sources, it separates OOA into factors with different degrees of oxygenation and thus presumed volatility, such as semi-volatile oxygenated organic aerosol (SV-OOA) and low-volatile oxygenated organic aerosol (LV-OOA; Ulbrich et al., 2009; Jimenez et al., 2009; Lanz et al., 2007). Abovementioned state-of-the-art instruments (e.g. Orbitrap, FIGAERO-HR-ToF-CIMS, EESI-ToF) enable the link between the molecular composition of OA and its physicochemical properties by use of molecular information in volatility parameterizations to calculate effective

saturation mass concentrations ($C_{sat}$) of different compounds (Li et al., 2016; Donahue et al., 2011). This can be used to define e.g. volatility basis sets (VBS), a framework that has been proposed and widely used for grouping the organic compounds into volatility classes (or bins) based on their $C_{sat}$ values (Donahue et al., 2006; 2011; 2012; Cappa and Jimenez, 2010).

Volatility determines whether an organic compound partitions into the particle phase and contributes to OA

particulate mass. It is thus an important physicochemical property of OA that influences the lifetime of OA and with that air quality. As a consequence of the connection between a compound's molecular composition and structure with its volatility, different types of OA fall into different categories of volatility. For e.g. OA measured in Mexico City, biomass burning OA (BBOA) was found to be the most volatile, followed by hydrocarbon-like OA (HOA), SV-OOA, and LV-OOA (Cappa and Jimenez, 2010). Isoprene epoxydiol (IEPOX) derived SOA was

observed to have the highest volatility of the OA measured in the southeastern U.S. (Lopez-Hilfiker et al., 2016). As ambient particles generally consist of a matrix of thousands of different compounds, OA apparent volatility can also be influenced by particle-phase diffusion limitations, e.g. due to amorphous phase state and/or the presence of a high mass fraction of oligomers (Vaden et al., 2011; Roldin et al., 2014; Cappa and Wilson, 2011; Yli-Juuti et al., 2017; Huang et al., 2018). Overall, the relationship between OA molecular composition and its volatility,

and how this relationship is influenced by environmental conditions and particle physicochemical properties, are not well characterized, particularly for field data.

Here we present detailed chemical composition measurements of OA from July–August 2017 and February–March 2018 in the city of Stuttgart, Germany. We investigate the molecular composition and volatility of OA particles, and discuss their seasonal variations as well as their potential sources. Stuttgart, a city located in

southwest Germany with a population of more than 600000 in a metropolitan area of 2.6 million inhabitants, is an important industrial center in Germany. It is situated in the steep valley of the Neckar river, in a "bowl" surrounded by a variety of hills, small mountains, and valleys. The complex topography can prevent the dispersion of air pollutants, and the location is characterized by low wind speeds and weak air circulation (Schwartz et al., 1991; Hebbert and Webb, 2012). Air quality has been a long-standing concern in Stuttgart, as it is one of the most polluted

cities in Germany (Schwartz et al., 1991; Süddeutsche Zeitung, 2016; Office for Environmental Protection, 2016); however only few detailed studies are available. For the year 2017, the state environmental protection agency, LUBW (Landesanstalt für Umwelt Baden-Württemberg), attributes 58 % of the annual mean $PM_{10}$ at their monitoring station "Am Neckartor" in downtown Stuttgart to road traffic (45 % abrasion, 7 % exhaust, 6 % secondary formation), 8 % to small and middle size combustion sources, and 27 % to regional background (LUBW,

2019). Mayer (1999) showed the temporal variability of urban air pollutants (NO, $NO_2$, $O_3$, and $O_x$ (sum of $NO_2$ and $O_3$)) caused by motor traffic in Stuttgart based on more than 10 years of observations, with higher NO concentrations in winter and higher $O_x$ concentrations in summer. Bari et al. (2011) characterized air pollutants such as polycyclic aromatic hydrocarbons (PAHs) and other wood smoke tracer compounds (levoglucosan, methoxyphenols) from wood-burning in the residential areas of Dettenhausen (about 30 km south of Stuttgart) and





attributed 57% of the ambient $PM_{10}$ pollution to hardwood combustion during wintertime. Our study therefore adds an important piece of information on air quality in Stuttgart by investigating the chemical composition, physicochemical properties, and potential sources of the OA particles in this city.

## 2 Methodology

### 2.1 Measurement site

We performed particle and trace gas measurements from July 5th to August 17th, 2017 and from February 5th to March 5th, 2018 in the city of Stuttgart, Germany (48°47'55.1"N, 9°12'13.5"E). The measurement site was located near the park "Unterer Schlossgarten" of Stuttgart and can be classified as an urban background site. The only nearby source is a parcel distribution center with delivery trucks passing by with low frequency during daytime. It was set up on a bridge over a train track about 2.2 km northeast of the Stuttgart main station with frequent train

traffic (electric). The air quality monitoring station of LUBW, "Am Neckartor", is 1.5 km southwest and one of the busiest roads in Stuttgart, B14 (LUBW, 2019), is about 360 m southwest of the measurement location.

All instruments were set up in a temperature-controlled measurement container kept at ~298 K. The container has been described elsewhere (Huang et al., 2019; Shen et al., 2018). All sampling inlets were located 3.7 m above ground level and 1.5 m above the container roof. An overview of instruments and parameters measured is given in

Table S1 in the Supplement.

### 2.2 Meteorological, particle and trace gas measurements

Temperature, relative humidity, wind direction, wind speed, global radiation, pressure, and precipitation data were measured by a meteorological sensor (WS700, Lufft GmbH; see Table S1). The main wind directions during the campaign were southwest during the summer and northeast during the winter. Trace gases ($O_3$, $CO_2$, $NO_2$, $SO_2$)

were measured with the corresponding sensors (Table S1). Particle number concentrations were recorded with two condensation particle counters (a CPC3022, measuring particles with mobility diameters larger than 7 nm, and a CPC3776, measuring particles with mobility diameters larger than 2.5 nm, both TSI Inc.). Particle size distributions were measured with a nanoscan scanning mobility particle sizer (NanoScan SMPS3910, measuring particles with mobility diameters between 10 nm and 420 nm, TSI Inc.). Black carbon (BC) concentrations were measured with

an Aethalometer (AE51, Aethlabs Inc.).

A high-resolution time-of-flight aerosol mass spectrometer (HR-ToF-AMS, Aerodyne Research Inc., hereafter AMS) equipped with an aerodynamic high-pressure lens (Williams et al., 2013) was deployed to continuously measure total non-refractory particle mass as a function of size (up to 2.5 μm particle aerodynamic diameter $d_{va}$) at a time resolution of 0.5 min. The AMS inlet was connected to a $PM_{2.5}$ head (flow rate 1 $m^3$ $h^{-1}$, Comde-Derenda

GmbH) and a stainless steel tube of 3.45 m length (flow rate 0.1 L $min^{-1}$, residence time 0.9 s). AMS data were analyzed with the AMS data analysis software packages SQUIRREL (version 1.60C) and PIKA (version 1.20C). Polytetrafluoroethylene (PTFE) filters (Zefluor PTFE membrane, 2 μm pore size, 25 mm diameter, Pall Corp.), which were prebaked at 200 °C in an oven overnight and stored in clean filter slides, were deposited during daytime (between 10:00 and 16:00) using a stainless steel filter holder connected to a $PM_{10}$ head (flow rate 1 $m^3$ $h^{-1}$, Comde-

Derenda GmbH) via a stainless steel tube and conductive tubing of 2.85 m length (flow rate 8.7 L $min^{-1}$ (summer) or 10 L $min^{-1}$ (winter), residence time 0.75 s (summer) or 0.72 s (winter)). Deposition times were varied (20–260



min) based on ambient organic mass concentrations in order to achieve similar mass concentrations deposited on the filter to avoid mass loading effects (Huang et al., 2018; Wang and Ruiz, 2018). A total of 21 filter samples were collected in the summer and 10 in the winter. After deposition, each filter sample was stored in a filter slide,

wrapped in aluminium foil, and then stored in a freezer at −20 °C until analysis in the laboratory by a filter inlet for gases and aerosols coupled to a high-resolution time-of-flight chemical ionization mass spectrometer (FIGAERO-HR-ToF-CIMS, Aerodyne Research Inc., hereafter CIMS) deploying iodide (I⁻) as reagent ion. Particles collected on the filter were thermally desorbed by a flow of ultra-high purity (99.999 %) nitrogen heated to 200 °C over the course of 35 min. The resulting mass spectral desorption signals are termed thermograms

(Lopez-Hilfiker et al., 2014). For individual compounds, signals that peak at distinct desorption temperatures ($T_{max}$) correlate with their saturation vapor pressure (Lopez-Hilfiker et al., 2015; Mohr et al., 2017); however, interference from isomers with different vapor pressures or thermal fragmentation of larger oligomeric molecules can lead to more complex, multimodal thermograms (Lopez-Hilfiker et al., 2015). Integration of thermograms of individual compounds yields their total particle-phase signal. We assume the sensitivity to be the same for all compounds

measured by CIMS (Huang et al., 2019) and convert the signal to mass so that the molecular weight of a compound is taken into account. In this study we do not attempt to derive any atmospheric mass concentrations from these filter measurements, since the actual deposited area of aerosol particles on the filter was larger than the area of the desorption flow, and the deposition was not evenly distributed across the filter. We therefore focus on the molecular composition and volatility distributions of OA particles, their seasonal variations, and the interpretation of these

observations for potential sources. In order to correct for filter backgrounds, we collected prebaked clean filters from the measurement site without deposition flow for both winter and summer. Field blank samples for winter and summer were analyzed by CIMS in the laboratory and used for background subtraction.

### 3 Results and discussion

#### 3.1 Particulate OA mass loadings

We observe higher total non-refractory $PM_{2.5}$ mass concentrations measured by AMS in winter (27.0 ± 11.9 µg m⁻³, average ± 1 standard deviation) than in summer (7.1 ± 3.3 µg m⁻³) at this measurement site (Figure S1). Similar observations were also made for other central European locations, e.g. Zurich, Switzerland (Jimenez et al., 2009; Zhang et al., 2007). Reasons for this observation are differences in emission sources between the seasons, boundary layer height dynamics, and/or meteorological conditions (Canonaco et al., 2015; Daellenbach et al., 2019;

Baumbach and Vogt, 2003). The surface inversion, which develops by radiative cooling of the ground and is dissolved from the bottom up by solar radiation and heating up of the ground (Baumbach and Vogt, 2003), is expected to be stronger in winter due to lower ambient temperature and global radiation (Figure S2), weaker air circulation (i.e. wind speed, Figure S3), and less precipitation. Air pollutants are therefore more likely to be kept beneath this inversion and have longer local residence time in the atmosphere in winter. While the average organic

mass loadings measured by AMS are lower in summer (5.1 ± 3.2 µg m⁻³) than in winter (8.4 ± 5.6 µg m⁻³; see Fig. S1), organics contribute relatively more mass to total non-refractory $PM_{2.5}$ measured by AMS in summer (68.8 ± 13.4 %) compared to winter (34.8 ± 9.5 %; see Fig. S1). Contributions of fragments containing only C and H atoms (CH) or also one oxygen atom ($CHO_1$) or more than one oxygen atoms ($CHO_{gt1}$) to total OA measured by AMS are similar, with slightly higher contributions of CH and $CHO_1$ in summer (CH: 29.4 ± 3.9 %; $CHO_1$: 15.7 ± 1.6



%) compared to winter (CH: 27.9 ± 4.6 %; CHO$_1$: 15.3 ± 1.9 %) and slightly lower contributions of CHO$_{gt1}$ in

summer (14.0 ± 2.6 %) compared to winter (15.8 ± 2.8 %). This is also reflected in the higher elemental oxygen-

to-carbon (O:C) ratios measured by AMS in winter (0.61 ± 0.12) than in summer (0.55 ± 0.10). The results imply

that OA is more oxygenated in winter. Due to fragmentation of organic molecules during the ionization process in

the AMS, molecular information of OA is lost. This information is able to retrieve from the filter samples analyzed

by CIMS. Due to the fact that the iodide CIMS is selective towards polarizable and thus oxygenated compounds

(Lee et al., 2014), the organic compounds measured by CIMS are oxygenated organic aerosol (OOA). In the next

section we will discuss the molecular composition of OOA measured by CIMS.

**3.2 Molecular composition of OOA**

Figure 1a and 1c show a comparison of CIMS mass spectral patterns of all CHOX compounds (C$_{x≥1}$H$_{y≥1}$O$_{z≥1}$X$_{0-n}$

detected as clustered with I$^-$, with X being different atoms like N, S, Cl, or a combination thereof; 1808 out of a

total of 2138 identified compounds and accounting for >96 % of total signals), and CHON compounds only for

summer and winter (panels b, d). Mass spectra shown were normalized to the sum of the deposited mass of all

detected CHOX compounds. Although the absolute CHOX mass concentrations are uncertain, the time series of

the sum of the deposited mass of all detected CHOX compounds follows the trend of the OA concentrations

measured by AMS quite well. CHO compounds (compounds containing only C, H, and O atoms) are the

dominating group and make up 79.4 ± 3.3 % of total CHOX in summer and 74.6 ± 2.2 % of total CHOX in winter,

followed by CHON compounds with 20.1 ± 3.4 % of total CHOX in summer and 24.6 ± 2.4 % of total CHOX in

winter. CHON compounds contribute relatively more mass in winter (also reflected in the organic bound nitrate

fraction (OrgNO$_3$, i.e., organonitrates) from AMS data (summer: 0.3 ± 0.2 µg m$^{-3}$; winter: 1.7 ± 1.1 µg m$^{-3}$),

determined assuming an NO$_2^+$/NO$^+$ ratio of OrgNO$_3$ of 0.1; Farmer et al., 2010; Kiendler-Scharr et al., 2016; see

Figure S4 and S5), while CHO compounds contribute relatively more mass in summer. This is possibly due to the

higher daytime O$_3$ concentrations in summer and higher daytime NO$_2$ concentrations in winter (Figure S6) as well

as different emission sources. Contributions of some biogenic marker compounds are higher in summer (Fig. 1a–

b), particularly C$_8$H$_{12}$O$_5$ (molecular formula corresponding to 2-hydroxyterpenylic acid identified in α-pinene SOA

by Claeys et al., 2009; Kahnt et al., 2014) and C$_8$H$_{11}$O$_7$N (identified in the laboratory as α-pinene oxidation product

by Lee et al., 2016). We also observe good correlations (Pearson's R: 0.85; Figure S7a) between our summer mass

spectra and the summer daytime mass spectra acquired in 2016 near Karlsruhe (a city in southwest Germany, about

70 km northwest of Stuttgart; Huang et al., 2019), indicative of the regional nature of sources and/or chemistry in

summer. We therefore conclude that the majority of the precursor VOC for OOA presented here in summer are

most likely of biogenic origin, despite the urban location of the measurement site.

Significantly higher contributions of C$_6$H$_{10}$O$_5$ (molecular formula corresponding to levoglucosan, a tracer for

biomass burning; Saarnio et al., 2010) are observed in winter compared to summer (Fig. 1c). Besides, higher

contributions of C$_6$H$_5$O$_3$N, C$_7$H$_7$O$_3$N, C$_6$H$_5$O$_4$N, and C$_7$H$_7$O$_4$N (molecular formulae corresponding to nitrated

phenols, tracers for biomass burning identified by Mohr et al., 2013) are also observed in winter (Fig. 1d). Some

of these compounds were also observed in the central European city of Zurich, Switzerland in winter (Daellenbach

et al., 2019). We cannot completely exclude that these compounds may have contributions from vehicular

emissions (Tong et al., 2016). However, significantly higher contributions of levoglucosan and nitrated phenols

indicate that biomass burning emissions are a dominant contributor to OOA in Stuttgart in winter. In addition to

compounds from biomass burning, we also observe similar patterns of contributions of CHON compounds with





 $m/z$ between 300–400 Th (also dominated by $C_8H_{11}O_7N$), and high contributions of $C_8H_{12}O_4$ (molecular formula corresponding to terpenylic acid identified in α-pinene SOA by Claeys et al., 2009) and $C_8H_{12}O_6$ (molecular formula corresponding to 3-methyl-1,2,3-butanetricarboxylic acid (MBTCA) in α-pinene SOA identified by Szmigielski et al., 2007; Müller et al., 2012) in winter. After removing the five biomass burning tracer compounds ($C_6H_{10}O_5$, $C_6H_5O_3N$, $C_7H_7O_3N$, $C_6H_5O_4N$, and $C_7H_7O_4N$), good correlations (Pearson's R: 0.70; Figure S7b) are

observed between summer mass spectra and winter mass spectra, indicating that biogenic emissions may also contribute significantly to the OOA particulate mass in winter. In addition, in both summer and winter, contributions of $C_7H_8O_5$ (identified in the laboratory as toluene oxidation product by Hinks et al., 2018; Molteni et al., 2018) are also observed, with relatively higher contributions in winter than in summer, indicating anthropogenic influences related to traffic or industrial activities (EPA, 1994).

In the following we will have a closer look at the bulk molecular composition for winter and summer daytime OOA measured by CIMS. Consistent with the O:C ratios measured by AMS (winter: 0.61 ± 0.12; summer: 0.55 ± 0.10), higher O:C ratios are also observed by CIMS in winter (0.89 ± 0.06) compared to summer (0.82 ± 0.02), despite lower ambient temperature and weaker global radiation in winter (see Fig. S2). The AMS O:C ratios are expected to be lower than those of the organic compounds measured by iodide CIMS, as the latter is selective

towards oxygenated compounds (Lee et al., 2014). Mass contributions of CHO and CHON with different number of oxygen atoms per molecule to total CHOX compounds as a function of the number of carbon atoms are shown in Figure 2. $C_8HO$ compounds exhibit the highest mass contributions in summer, while $C_6HO$ compounds surpass $C_8HO$ compounds in winter due to the large contributions of levoglucosan ($C_6H_{10}O_5$; see Fig. 2a and 2c). The mass distribution of CHO compounds in winter also exhibits higher contributions from compounds with 1–6 carbon

atoms and 4, 5 (levoglucosan), 7–9 oxygen atoms, while in summer higher contributions from compounds with 7–10 carbon atoms and 5–7 oxygen atoms are observed. This indicates that the slightly higher oxidation levels (or O:C ratios) in winter are related to both shorter carbon chain lengths and higher number of oxygen atoms of the OOA compounds compared to summer (see also Figure S8). In addition, relatively higher contributions of compounds with larger number of carbon atoms (C16–23) are also observed in winter (Fig. 2). A similar pattern

can be found for CHON compounds (Fig. 2b and 2d). $C_{9–10}HON$ compounds exhibit the highest mass contributions in summer, similar to what we observed in 2016 in summer near the city of Karlsruhe, where these compounds were determined to originate from biogenic VOC emissions (Huang et al., 2019). However, the filters in Stuttgart were deposited during daytime, therefore the chemistry involved in the formation of these CHON compounds likely involves the reaction of organic peroxy radicals ($RO_2$) with $NO_x$ instead of night-time $NO_3$ radical chemistry.

In winter, $C_6HON$ relative contributions exceed those from $C_{9–10}HON$ compounds, similar to the pattern of CHO compounds, indicative of similar sources (biomass burning emissions). Furthermore, in summer CHON compounds are dominated by compounds with 6–9 oxygen atoms, while in winter significantly higher contributions from compounds with 5–7 carbon atoms and 3–4 oxygen atoms are observed, mostly due to nitrated phenols ($C_{6–7}H_{5,7}O_{3–4}N$; see also Fig. 1d).

The results imply the importance of non-fossil OA formation from biogenic and/or biomass burning influences in different seasons even in a city with high traffic emissions mainly based on fossil fuel combustion (LUBW, 2019). This is similar to previous studies in other European cities such as Barcelona, Spain (Mohr et al., 2012) and some megacities in China (Ni et al., 2019). In the next section, we investigate the volatility of OOA compounds measured by CIMS, which can influence their lifetime in the atmosphere and thus air quality.



### 3.3 Seasonal changes of volatility of OOA

#### 3.3.1 Volatility distribution

Effective saturation mass concentrations ($C_{sat}$), a measure for volatility of a compound, were parameterized for each CHO and CHON compound using the approach by Li et al. (2016). The CHO and CHON compounds were then grouped into a 25-bin volatility basis set (VBS; Donahue et al., 2006) based on their $\log_{10}C_{sat}$ values (Figure 3). Organic compounds with $C_{sat}$ lower than $10^{-5}$ µg m$^{-3}$, between $10^{-4}$–$10^{-2}$ µg m$^{-3}$, and higher than $10^{-1}$ µg m$^{-3}$ are termed extremely low volatile organic compounds (ELVOC), low volatile organic compounds (LVOC), and semi-volatile organic compounds (SVOC), respectively. As shown in Fig. 3a, organic compounds with $C_{sat}$ between $10^2$–$10^3$ µg m$^{-3}$ make up the biggest mass contributions during daytime in both summer and winter, suggesting that SVOC is the dominating group in both seasons (summer: $74.2 \pm 3.4$ %; winter: $66.7 \pm 4.9$ %; see Fig. 3b–c). The dominating compounds in these volatility bins come from the group of $C_{8-12}HO$ compounds in summer and from the group of $C_{1-7}HO$ compounds with relatively higher O:C ratios in winter (Figure S9a and S9c). Dominant compounds are 2-hydroxyterpenylic acid ($C_8H_{12}O_5$) and levoglucosan ($C_6H_{10}O_5$) for summer and winter, respectively (compare also to Fig. 1a and 1c). Non-negligible contributions from $C_{1-7}HON$ compounds are also observed in these volatility bins in winter (Fig. S9b and S9d), mainly from nitrated phenols ($C_{6-7}H_{5,\ 7}O_{3-4}N$; compare also to Fig. 1d). In winter we also observe significant contributions of SVOC with $C_{sat}$ between $10^5$–$10^6$ µg m$^{-3}$ in the particle phase. Since winter is much colder compared to summer (Fig. S2), compounds of higher volatility are expected to be able to condense in winter. However, we may also have contributions from thermal decomposition products of oligomers to some of these low-molecular weight SVOC (also reflected in the multi-mode thermograms for CHOX compounds with 1–5 carbon atoms; see Figure S10). But for the larger compounds, such as dimers and trimers, contributions of thermal decomposition products become negligible (i.e. thermograms are unimodal; Huang et al., 2018; Wang and Ruiz, 2018). LVOC and ELVOC, which include compounds with larger molecular weight, exhibit higher mass contributions in winter (LVOC: $26.4 \pm 3.1$ %; ELVOC: $6.9 \pm 1.9$ %) than in summer (LVOC: $21.7 \pm 2.5$ %; ELVOC: $4.1 \pm 1.1$ %; see Fig. 3). The average mass-weighted $\log_{10}C_{sat}$ value is $1.05 \pm 0.28$ µg m$^{-3}$ for summer and $0.63 \pm 0.44$ µg m$^{-3}$ for winter.

The results indicate that even though the lower ambient temperatures in winter may lead to increased partitioning of SVOC into the particle phase, the bulk winter OOA is less volatile. Similar results were also observed in Zurich, Switzerland by Canonaco et al. (2015) based on AMS data. The lower volatility of OOA in Stuttgart in winter compared to summer can be partially explained by the higher contributions of compounds with shorter carbon chain lengths and higher number of oxygen atoms in winter (i.e. higher O:C ratios; see Fig. S8), and the relatively higher contribution of larger molecules (number of carbons atoms >16; see also Fig. 2). Differences in aging processes (functionalization, fragmentation, and oligomerization; Jimenez et al., 2009) between the seasons may also play a role, since Keller and Burtscher (2017) found that aging processes reduce the volatility of OA from biomass burning.

#### 3.3.2 Variation of the maximum desorption temperatures ($T_{max}$)

Thermograms resulting from the thermal desorption of the filter samples were analyzed. $T_{max}$, the maximum desorption temperatures at which the signals of a compound peak, were compared for summer and winter. Figure 4a–b shows the campaign-average high resolution two-dimensional (2D) thermogram of CHOX compounds, a



framework developed recently to investigate the OOA thermal desorption behavior over the entire $m/z$ and $T_{max}$
range (Wang and Ruiz, 2018). Each thermogram of each individual compound in the 2D space was normalized to
310   its maximum signal. Due to the CHOX compounds containing at least 1 carbon atom, 1 hydrogen atom, and 1
oxygen atom, and being detected as clustered with I⁻ ($m/z$ 126.9050 Th), the smallest $m/z$ in the 2D thermogram is
168 Th. As shown in Fig. 4a–b, the majority of the OOA compounds exhibit higher $T_{max}$ with a wider spread across
different CHOX compounds in winter (114.4 ± 17.1 °C, average ± 1 standard deviation) compared to summer
(96.8 ± 18.2 °C; see also Figure S11). For the summer period, $T_{max}$ decreases from 160 °C to 60 °C for $m/z$ 168–
315   280 Th, stays relatively constant at 60–110 °C for $m/z$ 280–550 Th, and increases from 60 °C to 120 °C for $m/z$ >550
Th; for the winter period, $T_{max}$ decreases from 160 °C to 80 °C for $m/z$ 168–280 Th, stays relatively constant at 2
$T_{max}$ regions (one region at 80–100 °C and the other one at 110–130 °C) for $m/z$ 280–550 Th, and increases from
80 °C to 130 °C for $m/z$ >550 Th (see also Fig. S11). The high $T_{max}$ values for $m/z$ <280 Th (SVOC range) result
from multi-mode thermograms with thermal decomposition of larger molecules (Lopez-Hilfiker et al., 2015). A
similar picture can be seen in the campaign-average thermograms for the sum of the signals of all CHOX, CHO,
and CHON compounds detected (normalized to the maximum; Fig. 4c). The pattern of the summer 2D thermogram
in Stuttgart, particularly the "zigzag"-like behavior of e.g. $m/z$ 280–380 Th (see Figure S12a), is comparable to
that from alkane-Cl SOA at high RH (67 %) and high $NO_x$ conditions observed by Wang and Ruiz (2018), and
was explained by increased hydroxyl functionalization over ketone functionalization. The winter 2D thermogram
in Stuttgart also has a "zigzag"-like pattern but less pronounced and at higher $T_{max}$ (see Fig. S12b).

The results indicate a generally lower apparent volatility (i.e. higher $T_{max}$) of bulk OOA in winter, in agreement
with the results based on the $C_{sat}$ parametrization (see also Fig. 3). Recent studies show that $T_{max}$ of a compound
can be influenced by isomers (Thompson et al., 2017), thermal fragmentation of larger molecules during the
heating of the filter (Lopez-Hilfiker et al., 2015), variations in filter mass loading (Huang et al., 2018; Wang and
Ruiz, 2018), and/or differences in particles' viscosity (Huang et al., 2018). Since deposited organic mass loadings
on the filter samples were similar for summer and winter, we can exclude a mass loading effect on the $T_{max}$ results
presented here. However, we also observed by eye that the filter samples in winter were more blackish, possibly
due to the higher black carbon (BC) concentrations during daytime (10:00–16:00) in winter (1247 ± 112 ng m⁻³)
compared to summer (1032 ± 311 ng m⁻³). If (and how) the higher BC concentrations can affect the desorption
behaviors of organic compounds (i.e. interactions between organic compounds and BC) is still unknown and
requires further laboratory studies. Higher O:C ratios (Buchholz et al., 2019), and/or higher oligomer mass
fractions (Huang et al., 2019; compare to LVOC and ELVOC mass contributions in Fig. 3b–c) have been shown
to be correlated with higher $T_{max}$, which is in agreement with our mass spectral observations in winter. Besides,
higher inorganic sulfate concentrations in winter (see Fig. S1) might also play a role in the formation of low volatile
but thermally unstable organic compounds which can only be detected as their decomposition products with
FIGAERO-CIMS (Gaston et al., 2016; Riva et al., 2019). If assuming that the winter thermograms are more
influenced by thermal decomposition of oligomers than the summer thermograms, which artificially shifts the
molecular formula-based volatility distribution towards higher volatility, the winter OOA are expected to be even
less volatile.

**4 Conclusions and atmospheric implications**





In this paper, chemical composition and volatility of OA particles were investigated during July–August 2017 and February–March 2018 in the city of Stuttgart, one of the most polluted cities in Germany. The average organic mass loadings measured by AMS are lower in summer ($5.1 \pm 3.2$ µg m$^{-3}$) than in winter ($8.4 \pm 5.6$ µg m$^{-3}$), but the relative contributions of OA to total non-refractory PM$_{2.5}$ mass measured by AMS are higher in summer ($68.8 \pm 13.4$ %) compared to winter ($34.8 \pm 9.5$ %). This can be explained by the differences in emission sources between the seasons, boundary layer height dynamics, and/or meteorological conditions (Canonaco et al., 2015; Daellenbach et al., 2019; Baumbach and Vogt, 2003). CIMS mass spectra from filter samples collected at the measurement site during daytime (10:00–16:00) show OOA compounds in summer have O:C ratios of $0.82 \pm 0.02$ and are more influenced by biogenic emissions (as shown by e.g. tracers of α-pinene oxidation products), while OOA compounds in winter have slightly higher O:C ratios ($0.89 \pm 0.06$) and are more influenced by biomass burning emissions (as shown by e.g. signals of levoglucosan and nitrated phenols).

The apparent volatility of the OOA compounds varies between the two seasons. OOA in winter is found to be less volatile, which is reflected in the higher contributions of LVOC and ELVOC in the VBS, as well as in the higher maximum desorption temperatures (T$_{max}$) of the organic compounds desorbing from the particles deposited on the filter samples. Potential possible reason for the lower apparent volatility of winter OOA is the increased residence time of air masses over Stuttgart due to the stronger surface inversion and thus longer atmospheric aging time of the OOA compounds, leading to a reduction in volatility (Keller and Burtscher, 2017; Jimenez et al., 2009). This is also consistent with the higher O:C ratios and the higher mass contributions of larger molecular-weight LVOC and ELVOC in winter. Since the OOA observed in the winter period also shows influence from biogenic emissions, the sources for the LVOC and ELVOC may partly be biogenic. In addition, interactions between different species and/or particles (particle matrix; Huang et al., 2018) due to higher BC, OA and inorganic concentrations, such as the intermolecular interactions between biomass burning compounds and biogenic/anthropogenic organic compounds and/or the interactions between organic compounds, inorganic compounds, and BC, might also play a role in the reduction of volatility of aerosol particles in winter. Overall, the lower apparent volatility of the winter OOA compounds could be caused by higher O:C ratios, but may also be related to the higher particle viscosity due to the higher mass contributions of larger molecular-weight LVOC and ELVOC, interactions between different species and/or particles deposited on the filter (particle matrix; Huang et al., 2018), and/or thermal decomposition of large molecules.

The results suggest that whereas lower temperatures in winter may lead to increased partitioning of SVOC into the particle phase, this does not result in a higher overall volatility of OOA in winter, and that the difference in sources and/or chemistry between the seasons plays a more important role. Our study provides insights into the seasonal variation of molecular composition and volatility of ambient OA particles during daytime, and into their potential sources, which is important for air pollution mitigation in urban locations. Our study shows the important contributions of non-fossil OA from biogenic and biomass burning even in an urban area with high traffic emissions mainly based on fossil fuel combustion (LUBW, 2019). As a consequence, in addition to mitigating traffic emissions, reducing emissions of anthropogenic OOA precursors from e.g. industry and biomass burning may contribute to reducing the environmental and health effects of air pollution.

**Data availability**

Data are available upon request to the corresponding author.



**Author contributions**


WH operated AMS and took the filter samples during the two field campaigns, analyzed the filters by CIMS in the laboratory, did the CIMS and AMS data analysis, produced all figures, and wrote and edited the manuscript; HS organized the campaign, did the trace gas, CPC, and black carbon data analysis, and provided suggestions for the data interpretation and discussion; XS operated AMS and took the filter samples during the field campaigns; RR

took the filter samples during the summer campaign; TL gave general advice and comments for this manuscript; CM provided suggestions for the data analysis, interpretation, discussion, and edited the manuscript. All authors contributed to the final text.

**Competing interests**

The authors declare no conflict of interest.

**Acknowledgements**

Technical support by the staff at IMK-AAF, and financial support by China Scholarship Council (CSC) for Wei Huang and Xiaoli Shen, is gratefully acknowledged. Support by the Deutsche Bahn AG, the University of Stuttgart, and the partners of the project "Three-Dimensional Observation of Atmospheric Processes in Cities (3DO)" (uc2-3do.org/) is gratefully acknowledged.

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





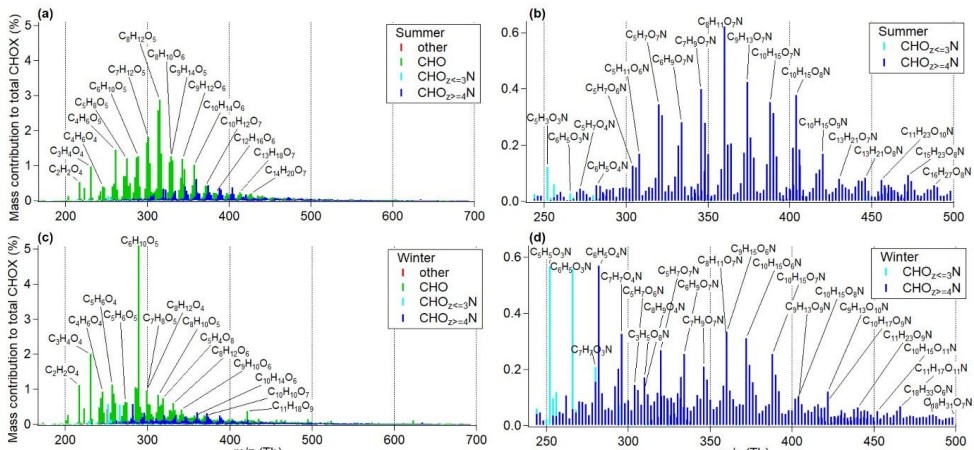

**Figure 1.** CIMS mass spectra comparison of CHOX compounds (separated into CHO, CHON and other compounds) (a), and CHON compounds (b) in the summer period, CHOX compounds (c) and CHON compounds (b) in the winter period as a function of *m/z* (includes mass of I⁻ ion; *m/z* 126.9050 Th). Mass contributions of each compound were normalized to the sum of the mass of all detected CHOX compounds.





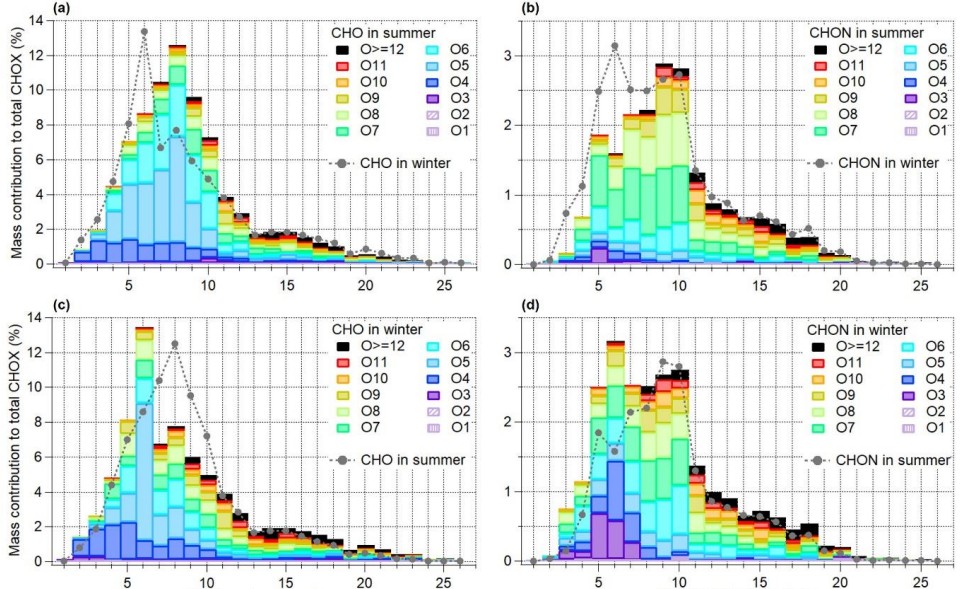

**Figure 2.** Mass contribution of CHO and CHON compounds with different number of oxygen atoms as a function of the number of carbon atoms to total CHOX compounds for the summer (a, c) and winter (b, d) periods. The corresponding distribution for the other season is plotted as a gray dotted line.



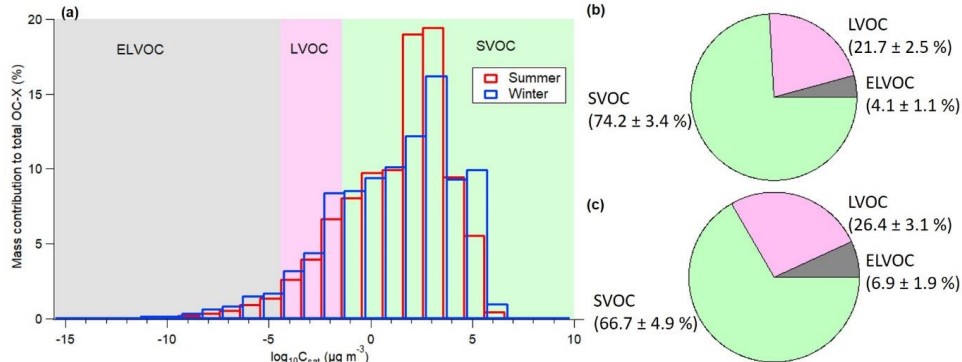


**Figure 3.** (a) Volatility distribution based on the molecular formula parameterization by Li et al. (2016); resulting

pie chart for the mass contributions of SVOC, LVOC, and ELVOC in the summer (b) and winter (c) periods.



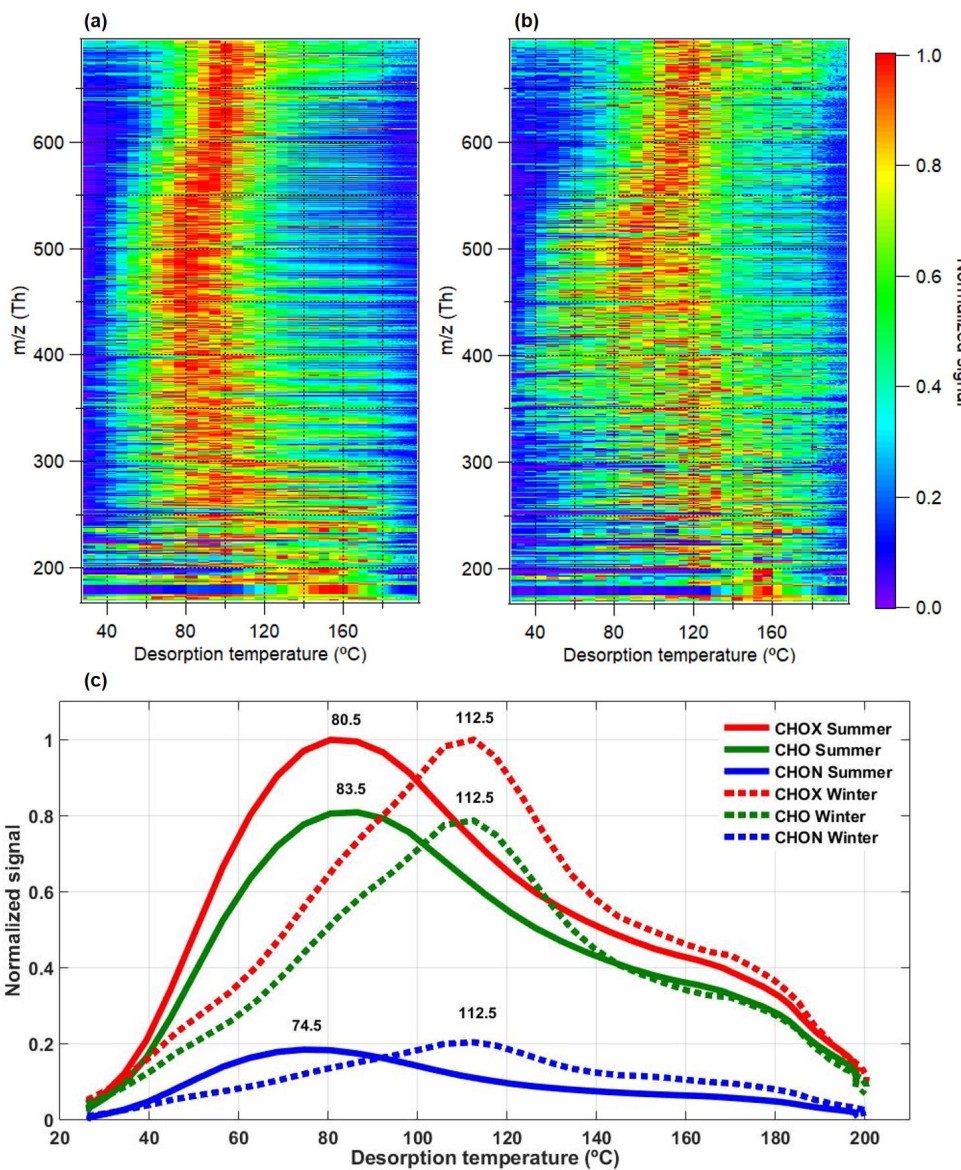

**Figure 4.** Comparison of campaign-average high resolution two-dimensional (2D) thermograms of CHOX compounds for the summer (a) and winter (b) periods vs $m/z$ (includes mass of I$^-$ ion; $m/z$ 126.9050 Th), and the sum thermograms of CHOX, CHO, and CHON compounds (c). The 2D thermograms and sum thermograms were normalized to their maximum values.