# Peer review of "Seasonal characteristics of organic aerosol chemical composition and volatility in Stuttgart, Germany"

_Atmospheric Chemistry and Physics, 2019_

## Referee Comment (RC1) · Anonymous Referee #1 · 28 May 2019

This manuscript presents seasonal differences in organic aerosol loading, chemical composition and volatility in Stuttgart, Germany using AMS and FIGAERO-CIMS measurements. They found that organic aerosols in the winter show lower volatilities and higher O:C compared to organic aerosols in the summer. Their dataset also provides information on sources of organic aerosols in the two seasons using identified species. Before this work is published in ACP, the authors need to provide careful clarification and further discussion of several important aspects in this manuscript. Please find the comments below.

General comments:

[Figure]

1. Were the filters that were collected in different seasons analyzed at different times of the year? If so, Tmax calibration using compounds with known vapor pressures may be required to constrain the instrument variability. It is possible that Tmax shifted for the same compounds due to differences in FIGAERO configuration and setup.

2. If I understand correctly, the filters were set up in the temperature-controlled room as well. Is it possible that in the winter campaign, when particles were sampled from the cold ambient air onto the filters held in the 298 K room, compounds with higher vapor pressures (probably SVOCs) already evaporated? If so, this will lead to under-estimation of the SVOC contribution in the winter.

Specific comments:

1. I suggest using O:C without the word "ratio" because the ":" means "ratio". The authors can just say "the oxygen to carbon ratio (O:C)" and subsequently just use O:C.

2. How similar are mass loadings for different filters? I suggest providing the mean and the standard deviation.

3. In line 187-191. How statistically different are the values in the Summer vs those in the Winter? It looks like they all fall within the uncertainty range.

4. In line 203, I suggest showing the time series plot of the OA concentration measured by the AMS versus the CHOX measured by the FIGAERO CIMS.

5. In line 258, although filters were deposited during daytime, the CHON compounds can come from NO3 oxidation from previous nights.

6. In line 273, I suggest presenting the volatility calculation here instead of just citing the reference.

7. In Figure 3, is there a reason why compounds with logC* > -1.5 are all labeled as SVOC? I suggest changing to the commonly-used volatility classes (SVOC: -0.5 < log10C* < 2.5; IVOC: -2.5 < log10C* < 6.5).

8. The paragraph starting in Line 141 is too long. I suggest breaking it into two or three shorter paragraphs.

---

## Referee Comment (RC2) · Anonymous Referee #2 · 11 Jun 2019

Huang et al. present a study investigating ambient organic aerosol in a German city with high air pollution for European standards. The combined information from two mass spectroscopy methods, AMS and FIGAERO-CIMS, gives insights into organic aerosol mass loadings, molecular composition, and apparent particle volatility. The study is well designed and enhances the current understanding of seasonal variability in the investigated organic aerosol properties. However, a few issues need to be addressed and discussed in the manuscript.

1.) The description of the filter collection suggests that the filters were collected inside the measurement container at ∼298K. This would be ∼25 °C warmer than the aver-

age ambient temperature in winter leading to a substantial evaporation of semi volatile compounds during sample collection even with the shortest collection times. The opposite could happen on hot summer days when the outside temperature is higher than in the container. Gas phase compounds would condense on the particles and filter increasing the observed volatile fraction. This needs to be discussed.

2.) It should be clearly stated in the text that the $C^*$ values from the elemental composition parameterization are calculated at a reference temperature. The "acting" $C^*$ values at the ambient temperature in summer and winter are different and with this the classification of compounds into EL/L/S/IVOC. It is possible to calculate the shift in $C^*$ values due to the different temperatures similar to Stolzenburg et al. 2018 and present them at least in the SI material.

3.) Generally, a little bit more information about the $C^*$ parameterization should be added. Especially a note that this assumes that each detected elemental composition is indeed only one isomer with one $C^*$ value and that no thermal decomposition occurred. The ion thermograms shown in Figure S10 indicate that this assumption is not universally valid, and you do discuss this later in the text. But in my opinion, this needs to be pointed out already when introducing the parameterization as it impacts the interpretation of the calculated $C^*$ values.

4.) The specific borders for the volatility categories vary between publications. But more resent ones (Donahue et al 2009) defines SVOC as -0.5 to 2.5 $\log_{10}C^*$ and IVOC from 2.5 to 6.5 $\log_{10}C^*$. Is there a reason for your different choice of categories?

5.) The dominant wind direction changes with the seasons from east to south-west coming over the inner city of Stuttgart including busy roads and the main train station with a big construction site. However, when discussing the seasonal changes in OOA sources this is not mentioned at all. Are the emissions so well mixed in that region that no influence on the SOA is expected?

Minor comments:

[Figure]

+ What were the mass loadings on the collected filters? Also, summer and winter samples had ∼20% different amounts of BC which is "invisible" to FIGAERO. Was the collected aerosol mass corrected for that?

+ In Figure S10, distinct changes in the ion thermograms are visible. However, due to the multitude of lines and the limited number of colours it is impossible to identify if e.g. any of the 3 dominant green lines in panel (a) are the same ion as the bimodal green line in panel (b). Adding the ion compositions as labels to a few ion thermograms may make reading this figure easier and may reveal some interesting details.

+ 1808 out of 2138 ions were of type CHOX. What were the other ones?

Technical comments:

page 2 line 63 "suggested" should be "suggesting"

page 4 line 127 "298 K" authors use Kelvin here and everywhere else temperature is given in Celsius. Should be changed to Celsius.

page 4 line 148 "filters were deposited" Particles are deposited on the filters, but the filters are not deposited in the filter holders.

page 5 line175 "m-3" is broken over lines

page 5 line189 "%" is broken in next line

page 10 line 360 "Potential possible reason" use either potential or possible.

References

Donahue et al., Atmos. Environ., 43(1), 94–106, doi:10.1016/j.atmosenv.2008.09.055, 2009.

Stolzenburg et al., Proc. Natl. Acad. Sci., 115(37), 9122–9127, doi:10.1073/pnas.1807604115, 2018.

---

## Author Comment (AC1) · 5 Aug 2019

*Responses to reviewers' comments for manuscript*

**Seasonal characteristics of organic aerosol chemical composition and volatility in Stuttgart, Germany**

Wei Huang[1,2], Harald Saathoff[1], Xiaoli Shen[1,2], Ramakrishna Ramisetty[1,a], Thomas Leisner[1,3], and Claudia Mohr[4]

[1]Institute of Meteorology and Climate Research, Karlsruhe Institute of Technology, Eggenstein-Leopoldshafen, 76344, Germany

[2]Institute of Geography and Geoecology, Working Group for Environmental Mineralogy and Environmental System Analysis, Karlsruhe Institute of Technology, Karlsruhe, 76131, Germany

[3]Institute of Environmental Physics, Heidelberg University, Heidelberg, 69120, Germany

[4]Department of Environmental Science and Analytical Chemistry, Stockholm University, Stockholm, 11418, Sweden

[a]now at: TSI Instruments India Private Limited, Bangalore, 560102, India

*Correspondence to: C. Mohr (claudia.mohr@aces.su.se)

*We thank all the reviewers for their evaluation of the manuscript, and for their constructive feedback. Replies to the individual comments are directly added below them in italics. Please note that only references that are part of the replies to the comments are listed in the bibliography at the end of this document. References in copied text excerpts from the manuscript are not included in the bibliography. Page and line numbers refer to the original manuscript text.*

**Reviewer #1** *(responses in italics)*

This manuscript presents seasonal differences in organic aerosol loading, chemical

composition and volatility in Stuttgart, Germany using AMS and FIGAERO-CIMS measurements. They found that organic aerosols in the winter show lower volatilities and higher O:C compared to organic aerosols in the summer. Their dataset also provides information on sources of organic aerosols in the two seasons using identified species. Before this work is published in ACP, the authors need to provide careful clarification and further discussion of several important aspects in this manuscript. Please find the comments below.

General comments:

1. Were the filters that were collected in different seasons analyzed at different times of the year? If so, Tmax calibration using compounds with known vapor pressures may be required to constrain the instrument variability. It is possible that Tmax shifted for the same compounds due to differences in FIGAERO configuration and setup.

*The filters, which were collected in different seasons, were analyzed at different times of the year. However, the instrument configuration and setup were kept exactly the same during these two measurements, and temperature is kept constant in our laboratory throughout the year. We therefore do not expect significant shifts in $T_{max}$ due to experimental setup. However, as shown recently (Huang et al., 2018; Wang and Ruiz, 2018), CIMS thermograms and corresponding $T_{max}$ can be influenced by different mass loadings on the filter. This was taken care of in our study by controlling the mass deposited on the filters by differing deposition times based on ambient concentrations of organic $PM_{2.5}$ measured concurrently by an AMS (compare lines 151-153 of the manuscript and specific comment 2 by reviewer 1 and minor comment 1 by reviewer 2).*

2. If I understand correctly, the filters were set up in the temperature-controlled room as well. Is it possible that in the winter campaign, when particles were sampled from the cold ambient air onto the filters held in the 298 K room, compounds with higher vapor pressures (probably SVOCs) already evaporated? If so, this will lead to underestimation of the SVOC contribution in the winter.

*For summer, the average ambient temperature during the deposition time (between 10:00 and 16:00) was 23.6 ± 1.7 °C, which is similar to the container temperature (25 °C). We therefore expect little effect of gaseous compounds condensing on or evaporating from the particles/filter during the summer period. For winter, the average ambient temperature during the deposition time (between 10:00 and 16:00) was lower*

*(2.3 ± 1.5 °C) than the container temperature (25 °C). We calculated a maximum temperature of ~19 °C for the air sample to reach potentially during deposition (Fitzer and Fritz, 1989). It is therefore possible that we will lose some highly volatile organic compounds during deposition in winter. For an attempt at characterizing these potential losses, we compared the CIMS mass spectra of chamber α-pinene SOA particles at room temperature (25 °C) and 5 °C (for both cases, the particle formation temperature and precursor concentrations were the same, and the CIMS was kept at room temperature; more information about the experimental setup can be found in Wagner et al. (2017) and Huang et al. (2018)). We find the mass spectra from these two experiments do not change significantly (Figure R1). In addition, thermograms (e.g. Figure 4) show the majority of organic compounds to start evaporating at desorption temperatures higher than 30 °C, with maximum desorption temperatures of >45 °C, even for SOA particles formed at -50 °C (Huang et al., 2018). We therefore assume that evaporation of highly volatile organic compounds during deposition in winter has a negligible influence on the main results of this study.*

[Figure]

*Figure R1. Scatter plot of particle phase signal for individual organic compounds of α-pinene SOA at 25 °C and 5 °C.*

Specific comments:

1. I suggest using O:C without the word "ratio" because the ":" means "ratio". The authors can just say "the oxygen to carbon ratio (O:C)" and subsequently just use O:C.

*Changed as suggested throughout the manuscript.*

2. How similar are mass loadings for different filters? I suggest providing the mean and the standard deviation.

*The organic mass deposited on the filter (summer: 3.5 ± 1.4 µg; winter: 4.0 ± 1.0 µg) was determined based on concurrent AMS measurements. This information was added to lines 152-153 of the manuscript: "[...] in order to achieve similar organic mass loadings on the filter (summer: 3.5 ± 1.4 µg; winter: 4.0 ± 1.0 µg based on concurrent AMS measurements during the deposition period) and to avoid mass loading effects [...]".*

3. In line 187-191. How statistically different are the values in the Summer vs those in the Winter? It looks like they all fall within the uncertainty range.

*We agree with the reviewer that the differences between these values fall within the uncertainty range. Therefore, the sentences in lines 187-193 were rephrased as following: "Contributions of fragments containing only C and H atoms(CH), or also one oxygen atom ($CHO_1$), or more than one oxygen atoms ($CHO_{gt1}$), to total OA measured by AMS are similar for both seasons (CH: 29.4 ± 3.9 % for summer and 27.9 ± 4.6 % for winter; $CHO_1$: 15.7 ± 1.6 % for summer and 15.3 ± 1.9 % for winter; $CHO_{gt1}$: 14.0 ± 2.6 % for summer and 15.8 ± 2.8 % for winter). Higher elemental oxygen-to-carbon ratios (O:C) measured by AMS were observed in winter (0.61 ± 0.12) than in summer (0.55 ± 0.10), implying that OA is more oxygenated in winter."*

4. In line 203, I suggest showing the time series plot of the OA concentration measured by the AMS versus the CHOX measured by the FIGAERO CIMS.

*As we mention in lines 166-168 and line 203 of the manuscript, we do not derive any atmospheric mass concentrations from these filter measurements, since the actual deposited area of aerosol particles on the filter was larger than the area of the desorption flow, and the deposition was not evenly distributed across the filter. Therefore, the absolute CHOX mass concentrations are uncertain. We have, however,*

*calculated the correlation coefficients between the OA concentrations measured by AMS and the CHOX measured by FIGAERO-CIMS (the latter in ug m⁻³, assuming a sensitivity of 22 cps ppt⁻¹, Lopez-Hilfiker et al., 2016; Huang et al., 2018), and they were added to lines 203-205 of the manuscript: "[...] the time series of the sum of the deposited mass of all detected CHOX compounds follows the trend of the OA concentrations measured by AMS quite well (Pearson's R: 0.95 for summer and 0.96 for winter)."*

5. In line 258, although filters were deposited during daytime, the CHON compounds can come from NO3 oxidation from previous nights.

*The sentence in lines 257-258 of the manuscript was rephrased as following: "The filters in Stuttgart were deposited during daytime, therefore the chemistry involved in the formation of these CHON compounds is likely dominated by the reaction of organic peroxy radicals ($RO_2$) with NOx; contributions of oxidation products formed via night-time $NO_3$ radical chemistry cannot be ruled out."*

6. In line 273, I suggest presenting the volatility calculation here instead of just citing the reference.

*Volatility calculation formula added as suggested.*

7. In Figure 3, is there a reason why compounds with logC* > -1.5 are all labeled as SVOC? I suggest changing to the commonly-used volatility classes (SVOC: -0.5 < log10C* < 2.5; IVOC: 2.5 < log10C* < 6.5).

*Based on reviewer 1's and reviewer 2's suggestion, we have added the IVOC category to Figure 3, and rephrased the corresponding text of the manuscript as following:*

*Lines 275-277: "Organic compounds with $C_{sat}$ lower than $10^{-4.5}$ μg m⁻³, between $10^{-4.5}$–$10^{-0.5}$ μg m⁻³, between $10^{-0.5}$–$10^{2.5}$ μg m⁻³, and between $10^{2.5}$–$10^{6.5}$ μg m⁻³ are termed extremely low volatile organic compounds (ELVOC), low volatile organic compounds (LVOC), semi-volatile organic compounds (SVOC), and intermediate volatile organic compounds (IVOC), respectively (Donahue et al., 2009)."*

*Lines 285-288: "In winter we observe much lower contributions of IVOC with $C_{sat}$ between $10^5$–$10^6$ μg m⁻³ in the particle phase. However, we may also have contributions from thermal decomposition products of oligomers to these low-molecular weight*

*compounds [...]".*

8. The paragraph starting in Line 141 is too long. I suggest breaking it into two or three shorter paragraphs.

*This paragraph was changed into two shorter paragraphs as suggested.*

**Reviewer #2** *(responses in italics)*

Huang et al. present a study investigating ambient organic aerosol in a German city with high air pollution for European standards. The combined information from two mass spectroscopy methods, AMS and FIGAERO-CIMS, gives insights into organic aerosol mass loadings, molecular composition, and apparent particle volatility. The study is well designed and enhances the current understanding of seasonal variability in the investigated organic aerosol properties. However, a few issues need to be addressed and discussed in the manuscript.

1.) The description of the filter collection suggests that the filters were collected inside the measurement container at ~298K. This would be ~25 °C warmer than the average ambient temperature in winter leading to a substantial evaporation of semi volatile compounds during sample collection even with the shortest collection times. The opposite could happen on hot summer days when the outside temperature is higher than in the container. Gas phase compounds would condense on the particles and filter increasing the observed volatile fraction. This needs to be discussed.

*Please see response to the general comment 2 of reviewer 1.*

2.) It should be clearly stated in the text that the C* values from the elemental composition parameterization are calculated at a reference temperature. The "acting" C* values at the ambient temperature in summer and winter are different and with this the classification of compounds into EL/L/S/IVOC. It is possible to calculate the shift in C* values due to the different temperatures similar to Stolzenburg et al. 2018 and present them at least in the SI material.

*We thank the reviewer for this suggestion. We have recalculated the VBS distributions*

*in Figure 3 and Figure S9 by taking the difference in ambient temperature between summer and winter into consideration. As a result, the VBS distribution for winter is shifted to lower volatility bins, making one of our main conclusions (winter OA particles are less volatile) even more clear. We have added/rephrased the corresponding text of the manuscript as following:*

*Lines 272-274: "[...] parameterized for each CHO and CHON compound using the approach by Li et al. (2016):*

$$log_{10} Csat (298 \; K) = (n_C^0 - n_C)b_C - n_O b_O - 2\frac{n_C n_O}{n_C + n_O}b_{CO} - n_N b_N \qquad (1)$$

*and then corrected for the summer (24 °C) and winter (2 °C) periods (Stolzenburg et al., 2018; Donahue et al., 2011; Epstein et al., 2010):*

$$log_{10} Csat (T) = log_{10} Csat (298 \; K) + \frac{\Delta H_{vap}}{Rln(10)}(\frac{1}{298} - \frac{1}{T}) \qquad (2)$$

$$\Delta H_{vap}(kJ \; mol^{-1}) = -5.7 \cdot log_{10} Csat (298 \; K) + 129 \qquad (3)$$

*[...]."*

*Lines 292-294: "[...] higher mass contributions in winter (LVOC: 37.0 ± 2.2 %; ELVOC: 15.9 ± 3.5 %) than in summer (LVOC: 22.6 ± 2.5 %; ELVOC: 4.8 ± 1.2 %; see Fig. 3b–c). The average mass-weighted log₁₀Csat value is 0.97 ± 0.28 µg m⁻³ for summer and -1.2 ± 0.48 µg m⁻³ for winter."*

3.) Generally, a little bit more information about the C* parameterization should be added. Especially a note that this assumes that each detected elemental composition is indeed only one isomer with one C* value and that no thermal decomposition occurred. The ion thermograms shown in Figure S10 indicate that this assumption is not universally valid, and you do discuss this later in the text. But in my opinion, this needs to be pointed out already when introducing the parameterization as it impacts the interpretation of the calculated C* values.

*The $C_{sat}$ parameterization was developed for bulk aerosol molecular composition based on volatility properties of functional groups (Donahue et al., 2011). When applied to individual molecules, with the only input being the molecular composition, isomers cannot be differentiated, as pointed out by the reviewer. It is also correct that the $C_{sat}$ parameterization cannot tell if a compound in the particle phase is a thermal*

*fragmentation product. We can only get this information when we look at e.g. the thermograms. To clarify these points, the following information was added to lines 273-274 of the manuscript: "We stress here that isomers cannot be differentiated with the $C_{sat}$ parameterization (Donahue et al., 2011) and that thermal fragmentation of organic compounds (Lopez-Hilfiker et al., 2015; Huang et al., 2018) during particle desorption with the FIGAERO can bias the $C_{sat}$ results towards higher volatilities. This will be discussed later. The CHO and CHON compounds were then grouped […]".*

4.) The specific borders for the volatility categories vary between publications. But more resent ones (Donahue et al 2009) defines SVOC as -0.5 to 2.5 log10C* and IVOC from 2.5 to 6.5 log10C*. Is there a reason for your different choice of categories?

*Based on reviewer 1's and reviewer 2's suggestion, we have added the IVOC category to Figure 3, and rephrased the corresponding text of the manuscript. The changes in the manuscript can be found in the response to specific comment 7 by reviewer 1.*

5.) The dominant wind direction changes with the seasons from east to south-west coming over the inner city of Stuttgart including busy roads and the main train station with a big construction site. However, when discussing the seasonal changes in OOA sources this is not mentioned at all. Are the emissions so well mixed in that region that no influence on the SOA is expected?

*We thank the reviewer for this input. We double-checked the mass contribution of the toluene oxidation product $C_7H_8O_5$, a marker compound for traffic emissions, as a function of wind direction for both seasons. Higher contributions are observed when the main wind direction is from the inner city of Stuttgart. The sentence in lines 238-239 of the manuscript was rephrased as following: "[…] with higher contributions in summer (when the main wind direction was from the inner city of Stuttgart, see Fig. S3) than in winter, indicating anthropogenic influences related to traffic or industrial activities (EPA, 1994)."*

Minor comments:

+ What were the mass loadings on the collected filters? Also, summer and winter samples had ~20% different amounts of BC which is "invisible" to FIGAERO. Was the collected aerosol mass corrected for that?

*Compare response to specific comment 2 by reviewer 1. The organic mass deposited on the filter (summer: 3.5 ± 1.4 μg; winter: 4.0 ± 1.0 μg) was determined based on concurrent AMS measurements.*

+ In Figure S10, distinct changes in the ion thermograms are visible. However, due to the multitude of lines and the limited number of colours it is impossible to identify if e.g. any of the 3 dominant green lines in panel (a) are the same ion as the bimodal green line in panel (b). Adding the ion compositions as labels to a few ion thermograms may make reading this figure easier and may reveal some interesting details.

*We thank the reviewer for this suggestion. We have added labels to the lines of a few dominant ion compositions to make this figure clearer.*

+ 1808 out of 2138 ions were of type CHOX. What were the other ones?

*The other ions are the reagent ion ($I^-$), inorganic compounds clustered with $I^-$, organic compounds and inorganic compounds not clustered with $I^-$, etc. Organic compounds not clustered with $I^-$ (289 compounds) were excluded in this analysis, as their ionization mechanism is highly uncertain. Assuming the same sensitivity as for the organic compounds detected as $I^-$-clusters, they only account for <0.03 % of the total particulate CHOX mass.*

Technical comments:

page 2 line 63 "suggested" should be "suggesting"

*We replaced "suggested" by "suggest".*

page 4 line 127 "298 K" authors use Kelvin here and everywhere else temperature is given in Celsius. Should be changed to Celsius.

*Done.*

page 4 line 148 "filters were deposited" Particles are deposited on the filters, but the filters are not deposited in the filter holders.

*Sentence rephrased as following: "Aerosol particles were deposited during daytime (between 10:00 and 16:00) on polyolytetrafluoroethylene (PTFE) filters (Zefluor PTFE*

*membrane, 2 μm pore size, 25 mm diameter, Pall Corp.) which were prebaked at 200 °C in an oven overnight and stored in clean filter slides, using a stainless steel filter holder [...]."*

page 5 line175 "m⁻³" is broken over lines

*Corrected.*

page 5 line189 "%" is broken in next line

*Corrected.*

page 10 line 360 "Potential possible reason" use either potential or possible.

*We replaced "Potential possible reason" by "Potential reason".*

**References:**

*Donahue, N. M., Epstein, S. A., Pandis, S. N., and Robinson, A. L.: A two-dimensional volatility basis set: 1. organic-aerosol mixing thermodynamics, Atmos Chem Phys, 11, 3303–3318, [https://doi.org/10.5194/acp-11-3303-2011](https://doi.org/10.5194/acp-11-3303-2011), 2011.*

*Fitzer, E., and Fritz, W.: Technische Chemie, Third ed., Springer, Berlin, 140 pp., 1989.*

*Huang, W., Saathoff, H., Pajunoja, A., Shen, X. L., Naumann, K.-H., Wagner, R., Virtanen, A., Leisner, T., and Mohr, C.: α-Pinene secondary organic aerosol at low temperature: chemical composition and implications for particle viscosity, Atmos Chem Phys, 18, 2883–2898, [https://doi.org/10.5194/acp-18-2883-2018](https://doi.org/10.5194/acp-18-2883-2018), 2018.*

*Lopez-Hilfiker, F. D., Iyer, S., Mohr, C., Lee, B. H., D'Ambro, E. L., Kurtén, T., and Thornton, J. A.: Constraining the sensitivity of iodide adduct chemical ionization mass spectrometry to multifunctional organic molecules using the collision limit and thermodynamic stability of iodide ion adducts, Atmos Meas Tech, 9, 1505−1512, [https://doi.org/10.5194/amt-9-1505-2016](https://doi.org/10.5194/amt-9-1505-2016), 2016.*

*Wagner, R., Höhler, K., Huang, W., Kiselev, A., Möhler, O., Mohr, C., Pajunoja, A., Saathoff, H., Schiebel, T., Shen, X. L., and Virtanen, A.: Heterogeneous ice nucleation of α-pinene SOA particles before and after ice cloud processing, J Geophys Res-Atmos, 122, 4924−4943, [https://doi.org/10.1002/2016JD026401](https://doi.org/10.1002/2016JD026401), 2017.*

*Wang, D. S., and Ruiz, L. H.: Chlorine-initiated oxidation of n-alkanes under high-NOx conditions: insights into secondary organic aerosol composition and volatility using*

*a FIGAERO-CIMS, Atmos Chem Phys, 18, 15535−15553, [https://doi.org/10.5194/acp-18-15535-2018](https://doi.org/10.5194/acp-18-15535-2018), 2018.*